# SePPO: Semi-Policy Preference Optimization for Diffusion Alignment

## Abstract

Reinforcement learning from human feedback (RLHF) methods are emerging as a way to fine-tune diffusion models (DMs) for visual generation. However, commonly used on-policy strategies are limited by the generalization capability of the reward model, while off-policy approaches require large amounts of difficult-to-obtain paired human-annotated data, particularly in visual generation tasks. To address the limitations of both on- and off-policy RLHF, we propose a preference optimization method that aligns DMs with preferences without relying on reward models or paired human-annotated data. Specifically, we introduce a Semi-Policy Preference Optimization (SePPO) method. SePPO leverages previous checkpoints as reference models while using them to generate on-policy reference samples, which replace "losing images" in preference pairs. This approach allows us to optimize using only off-policy "winning images." Furthermore, we design a strategy for reference model selection that expands the exploration in the policy space. Notably, we do not simply treat reference samples as negative examples for learning. Instead, we design an anchor-based criterion to assess whether the reference samples are likely to be winning or losing images, allowing the model to selectively learn from the generated reference samples. This approach mitigates performance degradation caused by the uncertainty in reference sample quality. We validate SePPO across both text-to-image and text-to-video benchmarks. SePPO surpasses all previous approaches on the text-to-image benchmarks and also demonstrates outstanding performance on the text-to-video benchmarks.

## 1 Introduction

Text-to-visual models have become a crucial component of the AIGC (AI-generated content) industry, with the denoising diffusion probabilistic model (DDPM) (Ho et al., 2020; Kingma et al., 2021) being the most widely used technology. However, current pre-trained text-to-visual models often fail to adequately align with human requirements. As a result, recent works (Liang et al., 2024a; Black et al., 2024; Wallace et al., 2024; Yang et al., 2024) adopt reinforcement learning (RL)-based approaches as the post-training process to better satisfy human needs, namely reinforcement learning from human feedback (RLHF).

These methods can generally be divided into two categories: on-policy methods and off-policy methods. Similar to large language models (LLMs), in the post-training of diffusion models, on-policy models use a reward model (RM) to score output and then backpropagate the policy gradients based on the scoring results. A typical approach is DDPO (Black et al., 2024), which utilizes vision language models as the reward model to improve the prompt-image alignment. Even though on-policy method is approved to be helpful in the Natural Language Processing (NLP) domain (Dong et al., 2024). However, aside from the existing issues with on-policy methods, such as reward hacking, which can lead to model collapse (Denison et al., 2024), in the text-to-visual task, despite the availability of numerous evaluation models, it is difficult to find a solution that can provide comprehensive feedback on all aspects of the visual content (Kim et al., 2024). Additionally, constructing an effective and efficient reward model is extremely challenging and heavily dependent on the collection of costly paired feedback data.

Another approach is to utilize a fixed set of preference data (generated by human or other models) as training data, which is called off-policy method. This method allows the trained model to achieve

a distribution similar to that of the preference data. An example is Diffusion-DPO (Wallace et al., 2024), which uses the Pick-a-Pic (Kirstain et al., 2023) dataset that contains paired preference image data generated by various models with human ratings. Apparently, off-policy methods also depend on human feedback data with positive and negative sample pairs, which requires additional efforts, and their results are usually inferior to those of on-policy methods (Tang et al., 2024). Thus, in this paper, our aim is to build a preference optimization method that can mitigate the issues of on-policy and off-policy, allowing the diffusion model to align with preferences without using a reward model or paired human-annotated data.

To enable models to learn preferences from human feedback, the construction of positive and negative examples is crucial. However, in most datasets, there is usually only one sample in each data point, which typically serves as a positive example. Therefore, in the absence of a reward model, our initial consideration is how to construct appropriate negative samples. Several previous works, such as SPIN-Diffusion (Yuan et al., 2024a) and DITTO (Shaikh et al., 2024), utilize the previous checkpoints to generate the so-called "losing" samples and then use preference optimization for model training. However, these methods cannot guarantee that the samples generated from the previous checkpoints are necessarily "losing" samples relative to the current model.

To address this issue, we propose a method called Semi-Policy Preference Optimization (SePPO). In our method, the positive samples are sampled from the supervised fine-tuning (SFT) dataset. In order to obtain sufficiently good negative samples that are not too far from the current model distribution, the "losing" samples are generated by the reference model. Unlike SPIN-Diffusion, where the reference model is set to use the latest checkpoint, and DITTO, where the reference model is the initial model. In our SePPO, the reference models are sampled from all the previous checkpoints. To be specific, we first study the selection strategies for the reference model, conducting experiments using three different approaches: (1) always selecting the initial checkpoint as the reference model, (2) selecting the checkpoint saved from the previous iteration, and (3) randomly selecting from all previous checkpoints. We found that as the number of training steps increases, compared to always using the initial checkpoint, selecting the checkpoint from the previous iteration makes the reference model less prone to overfitting and yields better results. Moreover, randomly selecting from all previous checkpoints produces similar results to selecting the checkpoint from the previous iteration and leads to a more stable training process overall.

In addition, to determine whether the samples generated from the reference model are genuinely "losing" images or "winning" images relative to the current model. We further design a strategy to determine whether the sampled examples are positive or negative samples, namely Anchor-based Adaptive Flipper (AAF). If we have a winning data point for the model to learn from and the reference model has a higher probability than the current model to generate this winning data point, then the probability of sampling a winning data point from the reference model distribution will be greater than that of generating a losing data point. In other words, in this case, the "losing images" are not truly negative samples for the current model. This could negatively affect the model if we continue to use the direct preference optimization (DPO) (Rafailov et al., 2024) or SPIN (Yuan et al., 2024a) loss functions. Therefore, we design a strategy where, if the reference model has a higher probability than the current model of generating the winning data point, the model will learn from both the winning data point and the samples generated by the reference model. This not only helps avoid the negative effects of incorrectly judging sample quality but also increases the chances of the model outperforming the results of SFT to some extent.

In summary, the main contributions of this work are as follows:

1. We design an iterative preference optimization method called Semi-Policy Preference Optimization (SePPO). Our model can achieve preference alignment without human annotation and a reward model, which reduces labor cost and infrastructure burden;

2. We develop a strategy that first samples the reference model in each iteration, which enables to expand the space of policy exploration. We then design a criterion to evaluate the quality of generated responses and adjust the preference optimization based on this quality information, a process termed Anchor-based Adaptive Flipper (AAF);

3. SePPO exceeds all previous optimization methods on the text-to-image benchmark, and its effectiveness has also been validated on the text-to-video datasets.

## 2 RELATED WORK

**Self-Improvement.** Self-improvement methods use iterative sampling to improve pre-trained models, which has the potential to eliminate the need for an expert annotator such as a human or more advanced models. A recent work INPO (Zhang et al., 2024) directly approximates the Nash policy (the optimal solution) of a two-player game on a human-annotated preference dataset. Concurrently, the demonstrated self-improvement methods (Shaikh et al., 2024; Chen et al., 2024) align pre-trained models only using supervised learning dataset (one demonstrated response for each prompt) without using reward models and human-annotated data pairs. Specifically, it takes the demonstrated response as the winner and generates the loser by reference models. DITTO (Shaikh et al., 2024) fixes the reference model as the initial model and works well on small datasets. SPIN (Chen et al., 2024) takes the latest checkpoint as the reference model and uses it to generate responses in each iteration. However, these approaches have the following shortcomings. First, they only focus on transformer-based models for the text modality. Second, the selection of reference models is fixed, which has limited space for policy exploration. Third, either a human-annotated preference dataset is necessarily required or they are built on a strong assumption that the demonstrated responses are always preferred to the generated responses. In fact, the responses generated by the models are not necessarily bad and do not always need to be rejected.

**RM-Free PO for Diffusion Model Alignment.** DPO-style methods (Rafailov et al., 2024) use a closed-form expression between RMs and the optimal policy to align pre-trained models with human preference. Thus, no RM is needed during the training process. Diffusion-DPO is first proposed in Wallace et al. (2024). Based on offline datasets with labeled human preference, it shows promising results on text-to-image diffusion generation tasks. Diffusion-RPO (Gu et al., 2024) considers semantic relationships between responses for contrastive weighting. MaPO (Hong et al., 2024) uses margin-aware information, and removes the reference model. However, only off-policy data is considered and human-annotated data pairs are necessarily required with high labor costs.

To involve on-policy data and minimize human annotation costs, self-improvement methods are being explored. SPIN-Diffusion (Yuan et al., 2024a) adapts the SPIN method to text-to-image generation tasks with diffusion models. It shows high efficiency in data utilization. However, first, it has the issues mentioned above as a self-improvement method. Second, only the text-to-image generation task is considered in all previous works.

## 3 BACKGROUND

### 3.1 DENOISING DIFFUSION PROBABILISTIC MODEL

Given a data sample $\mathbf{x}_0 \sim \mathcal{D}$, the forward process is a Markov chain that gradually adds Gaussian noise to the sample as follows

$$q(\mathbf{x}_{1:T}|\mathbf{x}_0) = \prod_{t=1}^{T} q(\mathbf{x}_t|\mathbf{x}_{t-1}), \quad q(\mathbf{x}_t|\mathbf{x}_{t-1}) = \mathcal{N}(\mathbf{x}_t|\sqrt{\alpha}\mathbf{x}_{t-1}, (1-\alpha)\mathbf{I}), \quad (1)$$

where $\mathbf{x}_1, \cdots, \mathbf{x}_T$ are latent variables, and $\alpha$ is a noise scheduling factor. Equivalently, $\mathbf{x}_t = \sqrt{\alpha}\mathbf{x}_{t-1} + \sqrt{1-\alpha}\epsilon$, where $\epsilon \sim \mathcal{N}(0, \mathbf{I})$.

With a latent variable $\mathbf{x}_t$ from the forward process, DDPM estimates the normalized additive noise by $\epsilon_\theta(\mathbf{x}_t)$, where $\theta$ represents the parameters of the neural network. To *maximize* the evidence lower bound (ELBO) (Kingma & Welling, 2019), we usually *minimize* the loss function w.r.t. $\theta$:

$$\mathcal{L}_{\text{DM}}(\theta; \mathbf{x}_0) = \mathbb{E}_{\epsilon,t}\left[w_t\|\epsilon_\theta(\mathbf{x}_t) - \epsilon\|^2\right], \quad (2)$$

where $t \sim \mathcal{U}(1, T)$ is uniformly distributed on the integer interval $[1, T]$, $w_t = \frac{(1-\alpha)^2\alpha^{t-1}}{2\sigma_t^2(1-\alpha^t)^2}$ is a weighting scalar, and $\sigma_t^2 = \frac{(1-\alpha)(1-\alpha^{\frac{t-1}{2}})}{1-\alpha^t}$ is the variance of the additive noise. $T$ as a prespecified constant in Kingma et al. (2021) is ignored in the loss function, because it has no contribution to training. In practice, $w_t$ is usually set to a constant (Song & Ermon, 2019).

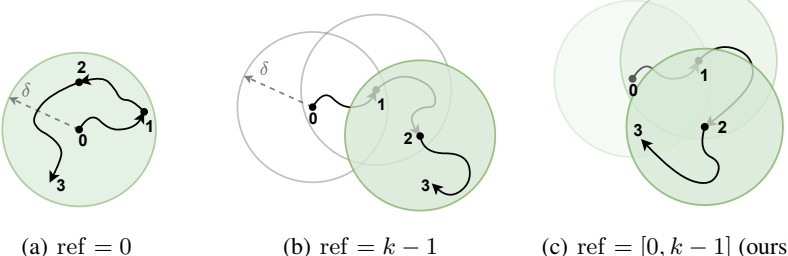

(a) ref $= 0$        (b) ref $= k - 1$        (c) ref $= [0, k - 1]$ (ours)

Figure 1: Illustration of the exploration area. In the $k$-th ($k = 3$) iteration, the green area satisfies the current KL constraint, while the blank area does not.

## 3.2 DIRECT PREFERENCE OPTIMIZATION

Given a prompt or condition $\mathbf{c}$, the human annotates the preference between two results as $\mathbf{x}^w \succ \mathbf{x}^l$. After preference data collection, in conventional RLHF, we train a reward function based on the Bradley-Terry (BT) model. The goal is to maximize cumulative rewards with a KullbackLeibler (KL) constraint between the current model $\pi_\theta$ and a reference model $\pi_{\text{ref}}$ (usually the initial model) as follows

$$\max_{\pi_\theta} \mathbb{E}_{\substack{\mathbf{c} \sim \mathcal{D}, \\ \mathbf{x} \sim \pi_\theta(\cdot|\mathbf{c})}} \Big[ r(\mathbf{x}, \mathbf{c}) - \beta D_{\text{KL}}\big(\pi_\theta(\mathbf{x}|\mathbf{c}) || \pi_{\text{ref}}(\mathbf{x}|\mathbf{c})\big) \Big], \tag{3}$$

where $r(\cdot)$ is the reward function induced from the BT model.

In DPO, the training process is simplified, and the target is converted to minimize a loss function as follows

$$\mathcal{L}_{\text{DPO}}(\theta) = \mathbb{E}_{(\mathbf{x}^w, \mathbf{x}^l, \mathbf{c}) \sim \mathcal{D}} \left[ - \log \sigma \Big( \beta \log \frac{\pi_\theta(\mathbf{x}^w|\mathbf{c})}{\pi_{\text{ref}}(\mathbf{x}^w|\mathbf{c})} - \beta \log \frac{\pi_\theta(\mathbf{x}^l|\mathbf{c})}{\pi_{\text{ref}}(\mathbf{x}^l|\mathbf{c})} \Big) \right], \tag{4}$$

where $\sigma(\cdot)$ (without subscript) is the logistic function.

# 4 ITERATIVE ALIGNMENT FOR DIFFUSION MODELS

## 4.1 ITERATIVE ALIGNMENT FROM SEMI-POLICY PREFERENCE OPTIMIZATION

When we have supervised fine-tuning data $(\mathbf{c}, \mathbf{x}^w)$, we consider using previous checkpoints to construct preference pairs. Specifically, in our method, we use the reference model for the sampling distribution to ensure high-quality generation of reference samples $\mathbf{x}^{\text{ref}}$. However, how to select an appropriate reference model remains a challenge.

In RL, equation 3 is a surrogate loss (Heess et al., 2017; Vieillard et al., 2020) of the hard KL-constrained target

$$\max_{\pi_\theta} \mathbb{E}_{\substack{\mathbf{c} \sim \mathcal{D}, \\ \mathbf{x} \sim \pi_\theta(\cdot|\mathbf{c})}} \Big[ r(\mathbf{x}, \mathbf{c}) \Big],$$

$$\text{s.t.} \quad \mathbb{E}_{\substack{\mathbf{c} \sim \mathcal{D}, \\ \mathbf{x} \sim \pi_\theta(\cdot|\mathbf{c})}} \Big[ D_{\text{KL}}\big(\pi_\theta(\mathbf{x}|\mathbf{c}) || \pi_{\text{ref}}(\mathbf{x}|\mathbf{c})\big) \Big] \leq \delta, \tag{5}$$

where the training path in the policy space (a family of parametric functions) can be visualized in a ball with radius $\delta$ as shown in Figure 1. $\delta$ as a hard constraint radius is roughly proportional to $1/\beta$. Assume that we save $K$ checkpoints in total during the training process. In the conventional DPO setting, the reference model is fixed to the initial checkpoint (ref $= 0$). However, this setting may have a limited and fixed exploration area controlled by the hyperparameter $\beta$. To better study the problem, in the $k$-th iteration, we design and test three sampling strategies for the reference model as follows:

- The reference model is sampled from the initial model, and is denoted as ref $= 0$.
- The reference model is sampled from the checkpoint saved in the last iteration, and is denoted as ref $= k - 1$.

---

**Algorithm 1** SePPO

---

**Require:** Demonstrated data set $(\mathbf{x}_0^w, \mathbf{c}) \sim \mathcal{D}$; Number of diffusion steps $T$; Number of iterations $K$; Initial model $\theta_0$.
1: **for** $k = 1, \cdots, K$ **do**
2:      Sample a reference model $\mathrm{ref} \sim \mathcal{U}(0, k-1)$.
3:      Generate $\mathbf{x}_0^{\mathrm{ref}}$ from $\theta_{\mathrm{ref}}$, and compose data pairs $(\mathbf{x}_0^w, \mathbf{x}_0^{\mathrm{ref}}, \mathbf{c})$.
4:      $\theta_k \leftarrow \theta_{k-1} - \eta_{k-1} \nabla \mathcal{L}(\theta; \mathbf{x}_0^w, \mathbf{x}_0^{\mathrm{ref}})$     # Or other optimizer, e.g., AdamW.
5: **end for**
**Ensure:** $\theta_K$

---

- The reference model is randomly sampled from all previously saved checkpoints, and is denoted as $\mathrm{ref} = [0, k-1]$.

The training algorithm is given in Algorithm 1. We omit $\mathbf{c}$ for the sake of concision. In each iteration, the loss function adapted from diffusion DPO is as follows

$$\mathcal{L}(\theta; \mathbf{x}_0^w, \mathbf{x}_0^{\mathrm{ref}}) = \mathop{\mathbb{E}}_{\epsilon^w, \epsilon^{\mathrm{ref}}, t} \Big[ -\log \sigma \Big( -\beta T w_t (\|\epsilon_\theta(\mathbf{x}_t^w) - \epsilon^w\|^2 - \|\epsilon_{\mathrm{ref}}(\mathbf{x}_t^w) - \epsilon^w\|^2$$
$$- \|\epsilon_\theta(\mathbf{x}_t^{\mathrm{ref}}) - \epsilon^{\mathrm{ref}}\|^2 + \|\epsilon_{\mathrm{ref}}(\mathbf{x}_t^{\mathrm{ref}}) - \epsilon^{\mathrm{ref}}\|^2) \Big) \Big],$$
(6)

where $t \sim \mathcal{U}(1, T)$, $\epsilon^w, \epsilon^{\mathrm{ref}} \stackrel{\mathrm{iid}}{\sim} \mathcal{N}(0, \mathbf{I})$, and $\mathbf{x}_t^{w,\mathrm{ref}} = \sqrt{\alpha^t} \mathbf{x}_0^{w,\mathrm{ref}} + \sqrt{1 - \alpha^t} \epsilon^{w,\mathrm{ref}}$. Notably, in equation 6, the reference image $\mathbf{x}_0^{\mathrm{ref}}$ is generated from the reference model $\theta_{\mathrm{ref}}$, which is on-policy learning.

To validate the effectiveness of $\mathrm{ref} = [0, k-1]$, we use the Pick-a-Pic dataset and the stable diffusion 1.5 (SD-1.5) model to conduct all experiments for the ablation study. In experiments, we save a checkpoint every 30 updates ($K = 7$ iterations). The experimental results are shown in Figure 2. In the beginning, the performances of all models are improved. However, as the number of steps $T$ increases, the performance of the model with $\mathrm{ref} = 0$ starts to decrease and quickly exhibits overfitting. The model with $\mathrm{ref} = k-1$ shows unstable performances. Due to the fact that when all reference samples are generated from the last checkpoint in the current round of training, the model is prone to fall into non-global optima, leading to instability in the training process Florensa et al. (2018). Therefore, considering both model performance and stability, our method uses $\mathrm{ref} = [0, k-1]$ to generate reference samples.

It is noteworthy that our positive samples $\mathbf{x}_0^w$ are collected off-policy from demonstrations, while the reference samples $\mathbf{x}_0^{\mathrm{ref}}$ are collected on-policy from the saved policies. Therefore, we term our approach Semi-Policy Preference Optimization (SePPO).

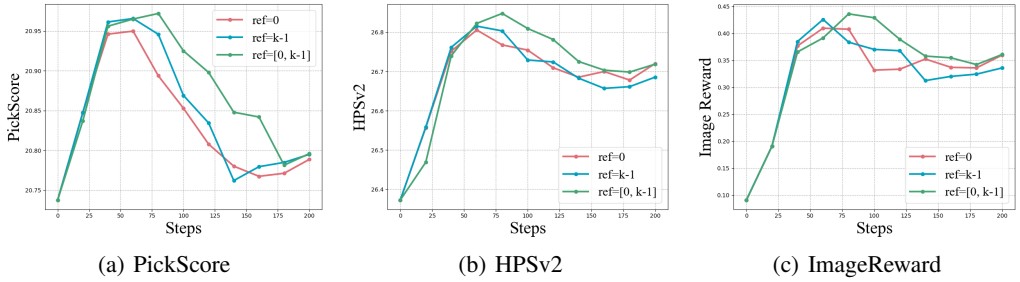

(a) PickScore        (b) HPSv2        (c) ImageReward

Figure 2: Ablation study of the reference model selection strategies. The x-axis is the number of diffusion steps $T$. The y-axis is the testing score. (The higher is better.)

### 4.2 Anchor-based Adaptive Flipper for Preference Optimization

On the other hand, this type of demonstrated method faces a significant problem: When using generated reference images $\mathbf{x}_0^{\mathrm{ref}}$, we cannot determine the relative relationship between the reference

model and the current model (whether the reference image is better than the image generated by the current model or not). Existing methods, when using a reference model to generate reference samples, always directly assume that the reference samples have relatively poor quality. For example, DITTO assumes that the earlier checkpoints used as the reference model produce worse-performing samples. However, as shown in the experiment in Figure 2, the reference model is not always inferior to the current model.

Specifically, based on the DPO-style optimization method, we analyze the change in gradient updates when the reference image is **better** than the image generated by the current model. The gradient of the loss function in equation 6 is

$$\nabla \mathcal{L}(\theta; \mathbf{x}_0^w, \mathbf{x}_0^{\text{ref}}) = \underset{\epsilon^w, \epsilon^{\text{ref}}, t}{\mathbb{E}} \left[ -2\beta T w_t \sigma \left( -\beta T w_t (\hat{\sigma}_{\text{ref}}^2 - \hat{\sigma}_w^2) \right) \underbrace{\left( \epsilon_\theta(\mathbf{x}_t^w) - \epsilon^w - \epsilon_\theta(\mathbf{x}_t^{\text{ref}}) + \epsilon^{\text{ref}} \right)}_{\text{error term}} \right],$$

(7)

where $\hat{\sigma}_w^2 := \|\epsilon_\theta(\mathbf{x}_t^w) - \epsilon^w\|^2 - \|\epsilon_{\text{ref}}(\mathbf{x}_t^w) - \epsilon^w\|^2$ and $\hat{\sigma}_{\text{ref}}^2 := \|\epsilon_\theta(\mathbf{x}_t^{\text{ref}}) - \epsilon^{\text{ref}}\|^2 - \|\epsilon_{\text{ref}}(\mathbf{x}_t^{\text{ref}}) - \epsilon^{\text{ref}}\|^2$. If $\mathbf{x}_t^{\text{ref}}$ generated by the reference model is good and close to $\mathbf{x}_t^w$, both the error term and the weight term $\sigma(\cdot)$ tend to be small. This leads to a very small gradient update without making full use of the information in the data sample, even though we know that $\mathbf{x}_0^w$ is generated by experts with high quality.

Therefore, in order to avoid the impact of the uncertainty of the reference image, we design a strategy that uses the winning image as an anchor, and evaluate the quality of the images generated by the reference model based on its performance relative to the current model on the anchor. Specifically, we design an Anchor-based Adaptive Flipper (AAF) to the loss function in equation 6 as follows:

$$\mathcal{L}(\theta; \mathbf{x}_0^w) = \underset{\epsilon^w, \epsilon^{\text{ref}}, t}{\mathbb{E}} \left[ -\log \sigma \left( -\beta T w_t \left( \|\epsilon_\theta(\mathbf{x}_t^w) - \epsilon^w\|^2 - \|\epsilon_{\text{ref}}(\mathbf{x}_t^w) - \epsilon^w\|^2 \right. \right. \right.$$
$$\left. \left. \left. - \text{sign} \cdot \left( \|\epsilon_\theta(\mathbf{x}_t^{\text{ref}}) - \epsilon^{\text{ref}}\|^2 - \|\epsilon_{\text{ref}}(\mathbf{x}_t^{\text{ref}}) - \epsilon^{\text{ref}}\|^2 \right) \right) \right) \right],$$

(8)

where $\epsilon^w, \epsilon^{\text{ref}} \overset{\text{iid}}{\sim} \mathcal{N}(0, \mathbf{I})$ and $\mathbf{x}_t^{w,\text{ref}} = \sqrt{\alpha^t} \mathbf{x}_0^{w,\text{ref}} + \sqrt{1 - \alpha^t} \epsilon^{w,\text{ref}}$. sign is a binary variable defined as

$$\text{sign} = \text{sgn}(\|\epsilon_{\text{ref}}(\mathbf{x}_t^w) - \epsilon^w\|^2 - \|\epsilon_\theta(\mathbf{x}_t^w) - \epsilon^w\|^2),$$

(9)

where $\text{sgn}(x) = 1$ if $x > 0$, and otherwise $\text{sgn}(x) = -1$.

Intuitively, if the reference model has a higher probability of generating noise $\epsilon^w$ compared to the current model, then in this situation, the reference image generated by the reference model is also more likely to be a winning image than a losing image. We formalize this claim in Theorem 4.1, which motivates the design of the criterion in equation 9.

**Theorem 4.1.** *Given two policy model parameters $\theta_1$ and $\theta_2$, it is almost certain that $\theta_1$ generates better prediction than $\theta_2$, if*

$$\mathbb{E} \left[ \|\epsilon_{\theta_1}(\mathbf{x}_t) - \epsilon\|^2 - \|\epsilon_{\theta_2}(\mathbf{x}_t) - \epsilon\|^2 \right] \leq 0,$$

(10)

*where $\epsilon \sim \mathcal{N}(0, \mathbf{I})$, and $\mathbf{x}_t = \sqrt{\alpha^t} \mathbf{x}_0 + \sqrt{1 - \alpha^t} \epsilon$.*

The proof of Theorem 4.1 is given in Appendix A.1. According to Theorem 4.1, if $\text{sign} = -1$, it means that $\mathbf{x}_0^{\text{ref}}$ generated by $\epsilon_{\text{ref}}$ has a high probability to be better than the output of $\epsilon_\theta$. In this situation, $\mathbf{x}_0^{\text{ref}}$ is of good quality and should not be rejected. The effectiveness of the AAF is further verified in the ablation study in Section 5.

## 5 EXPERIMENTS

### 5.1 SETUP

#### 5.1.1 TEXT-TO-IMAGE

In the text-to-image task, we test our methods based on the stable diffusion 1.5 (SD-1.5) model. Following Diffusion-DPO (Wallace et al., 2024), we use the sampled training set of the Pick-a-Pic

v2 dataset (Kirstain et al., 2023) as the training dataset. Pick-a-Pic dataset is a human-annotated preference dataset for image generation. It consists of images generated by the SDXL-beta (Podell et al., 2024) and Dreamlike models. Specifically, as mentioned in Diffusion-DPO, we remove approximately 12% pairs with ties and use the remaining $851,293$ pairs, which include $58,960$ unique prompts for training. We use AdamW (Loshchilov & Hutter, 2019) as the optimizer. We train our model on $8$ NVIDIA A100 GPUs with local batch size 1, and the number of gradient accumulation steps is set to 256. Thus, the equivalent batch size is 2048. We train models at fixed-square resolutions. A learning rate of $5 \times 10^{-9}$ is used, as we find that a smaller learning rate can help avoid overfitting. We set $\beta$ to 2000, which stays the same in Diffusion-DPO. For evaluation datasets, we use the validation set of the Pick-a-Pic dataset, the Parti-prompt, and HPSv2, respectively. We utilize the default stable diffusion inference pipeline from Huggingface when testing. The metrics we use are PickScore, HPSv2 score, ImageReward score, and Aesthetic score.

### 5.1.2 TEXT-TO-VIDEO

To further verify that SePPO works well in text-to-video generation tasks, we test our methods based on the AnimateDiff (Guo et al., 2024). We use the training set from MagicTime (Yuan et al., 2024b) as our training set and utilize the ChronoMagic-Bench-150 (Yuan et al., 2024c) dataset as our validation set. We use LoRA (Hu et al., 2022) to train all the models at the resolution $256 \times 256$ and 16 frames are taken by dynamic frames extraction from each video. The learning rate is set to $5 \times 10^{-6}$ and the training steps are set to 1000. The metrics we use are FID (Heusel et al., 2017), LPIPS (Zhang et al., 2018), SSIM (Wang et al., 2004), PSNR (Hore & Ziou, 2010) and FVD (Unterthiner et al., 2019).

## 5.2 RESULTS

### 5.2.1 ANALYSIS OF TEXT-TO-IMAGE

To verify the effectiveness of the proposed SePPO, we compare SePPO with the SOTA methods, including DDPO (Black et al., 2024), D3PO (Yang et al., 2024), Diffusion-DPO, SPO (Liang et al., 2024b) and SPIN-Diffusion. We first report all the comparison results on the validation unique split of Pick-a-Pic dataset in Table 1. Specifically, SFT$^w$ indicates that we use the **winning** images from the Pick-a-Pic dataset for supervised fine-tuning. SePPO$^r$ and SePPO$^w$ indicate that we use **randomly chosen** images or **winning** images in the Pick-a-Pic dataset as the training set for SePPO. SePPO outperforms previous methods across most metrics, even those that utilize reward models during the training process, such as DDPO and SPO. Moreover, SePPO does not require the three-stage training process like SPIN-Diffusion, nor does it require the complex selection of hyperparameters. Notably, we observe that SePPO significantly improves ImageReward, which may be attributed to the fact that ImageReward reflects not only the alignment between the image and human preference but also the degree of alignment between the image and the text. In contrast, the other metrics primarily reflect the alignment between the image and human preference.

Previous methods like SPIN-Diffusion used the winning images from the Pick-a-Pic dataset as training data. In our experiments, aside from training using the winning images as done previously, we also conduct an experiment where we randomly select images from both the winner and the losing sets as training data for SePPO, which we refer to as SePPO$^r$. Despite using a lower-quality training data distribution, SePPO$^r$ still outperforms other methods on most metrics. Notably, when compared to SFT$^w$, which is fine-tuned on the winning images, SePPO$^r$ still exceeds SFT$^w$ on three key metrics, further demonstrating the superiority of SePPO.

To better evaluate SePPO's out-of-distribution performance, we also test the model using the HPSv2 and Parti-prompt datasets, which have different distributions from the Pick-a-pic dataset. As shown in Table 2, SePPO outperforms all other models on these datasets. It is worth noting that SPO has not been tested on these two datasets and SPIN-Diffusion does not report their precise results. So, we reproduce the results by using the checkpoints of SPIN-Diffusion[1] and SPO[2] available on HuggingFace, referred to as SPIN-Diffusion* and SPO*.

On the left side of Figure 3, we further explore the relationship between our AAF rate and model performance. AAF rate is defined as the ratio $(\#\text{sign} = -1)/(\#\text{total})$ in a batch of data. The AAF

---

[1]https://huggingface.co/UCLA-AGI/SPIN-Diffusion-iter3

[2]https://huggingface.co/SPO-Diffusion-Models/SPO-SD-v1-5_4k-p_10ep

rate reflects the comparative performance between the current model and all previous checkpoints; a higher AAF rate indicates that the samples generated by the reference model are more likely to be negative samples for the current model, meaning that the current model is performing better relative to all previous models. We observe that, as training steps increase, both the AAF rate and PickScore gradually increase, showing a similar trend. We then display the changes in PickScore during the training process of SePPO and SFT on the right side of Figure 3. SFT quickly converges and begins to fluctuate, while SePPO is able to steadily improve throughout the training process.

We also visualize the results of SD-1.5, SPO, SPIN-Diffusion, and SePPO in Figure 4. SePPO is able to capture the verb "nested" and also generating better eye details. SePPO not only demonstrates superior visual quality compared to other methods but also excels in image-text alignment. This is because SePPO's training approach, which does not rely on a reward model to guide the learning direction, allows the model to learn both human preferences and improve in areas that reward models may fail to address.

Table 1: Model Feedback Results on the Pick-a-Pic Validation Set.

| Methods | PickScore ↑ | HPSv2 ↑ | ImageReward ↑ | Aesthetic ↑ |
|---|---|---|---|---|
| SD-1.5 | 20.53 | 23.79 | -0.163 | 5.365 |
| SFT[w] | 21.32 | 27.14 | 0.519 | 5.642 |
| DDPO | 21.06 | 24.91 | 0.082 | 5.591 |
| D3PO | 20.76 | 23.91 | -0.124 | 5.527 |
| Diffusion-DPO | 20.98 | 25.05 | 0.112 | 5.505 |
| SPO | 21.43 | 26.45 | 0.171 | 5.887 |
| SPIN-Diffusion[*] | 21.55 | 27.10 | 0.484 | **5.929** |
| SePPO[r] | 21.33 | 27.07 | 0.524 | 5.712 |
| SePPO[w] | **21.57** | **27.20** | **0.615** | 5.772 |

Table 2: Model Feedback Results on the HPSv2 and Parti-prompt Datasets.

| Methods | PickScore ↑ | | HPSv2 ↑ | | ImageReward ↑ | | Aesthetic ↑ | |
|---|---|---|---|---|---|---|---|---|
| | HPS | Parti | HPS | Parti | HPS | Parti | HPS | Parti |
| SD-1.5 | 20.95 | 21.38 | 27.17 | 26.70 | 0.08 | 0.16 | 5.55 | 5.33 |
| SFT[w] | 21.50 | 21.68 | 27.88 | 27.40 | 0.68 | 0.56 | 5.82 | 5.53 |
| Diffusion-DPO | 21.40 | 21.63 | 27.23 | 26.93 | 0.30 | 0.32 | 5.68 | 5.41 |
| Diffusion-RPO | 21.43 | 21.66 | 27.37 | 27.05 | 0.34 | 0.39 | 5.69 | 5.43 |
| SPO[*] | 21.87 | 21.85 | 27.60 | 27.41 | 0.41 | 0.42 | 5.87 | 5.63 |
| SPIN-Diffusion[*] | 21.88 | 21.91 | 27.71 | 27.58 | 0.54 | 0.51 | **6.05** | **5.78** |
| SePPO[w] | **21.90** | **21.93** | **27.92** | **27.69** | **0.70** | **0.58** | 5.94 | 5.64 |

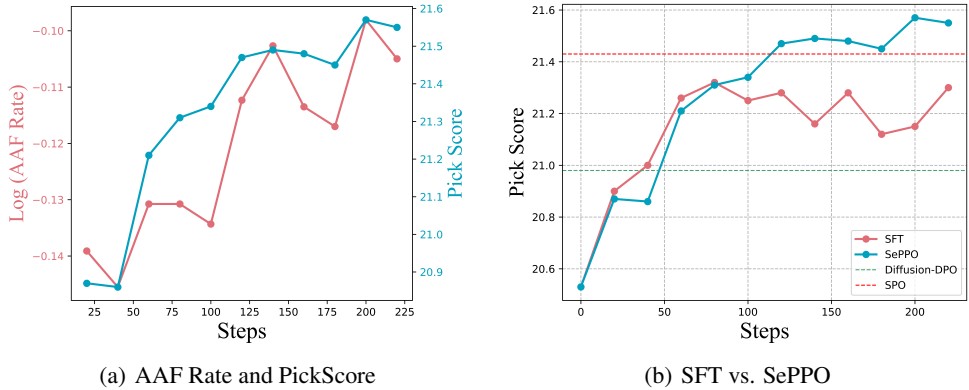

(a) AAF Rate and PickScore      (b) SFT vs. SePPO

Figure 3: Relations of AAF rates and model performance.

### 5.2.2 ANALYSIS OF TEXT-TO-VIDEO

We further validate SePPO on the text-to-video task in Table 3. Time-lapse video generation task is chosed and all the model are trained based on AnimateDiff. Our method achieves improvements

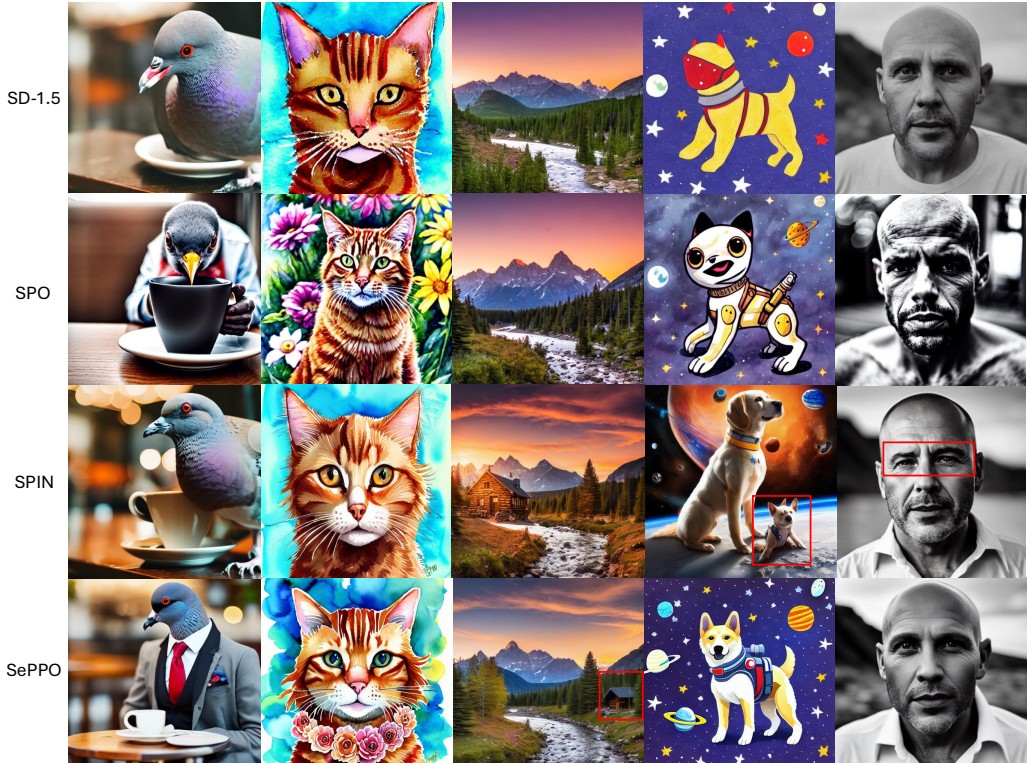

Figure 4: Text-to-image generation results of SD-1.5, SPO, SPIN-Diffusion and SePPO. The prompts from left to right are: (1) *Photo of a pigeon in a well tailored suit getting a cup of coffee in a cafe in the morning*; (2) *Ginger Tabby cat watercolor with flowers*; (3) *An image of a peaceful mountain landscape at sunset, with a small cabin nestled in the trees and a winding river in the foreground*; (4) *Space dog*; (5) *b&w photo of 42 y.o man in white clothes, bald, face, half body, body, high detailed skin, skin pores, coastline, overcast weather*.

across all metrics compared to both the vanilla and the fine-tuned AnimateDiff. We also visualize the video results in Figure 5, showing that SePPO achieves higher alignment with the text and better realism compared to the other methods.

### 5.2.3 ABLATION STUDY

We perform an ablation study of our method in Table 4. When we remove AAF, meaning that all reference samples are treated as negative samples, the model's performance drops significantly, demonstrating the effectiveness of AAF. Furthermore, when we replace the sign function $\text{sgn}(x)$ in equation 9 with the indicator function $\mathbb{1}(x > 0)$, *i.e.*, choosing to only learn from the winning image when the reference image has a higher probability of not being a losing image, and applying DPO when the reference image has a higher probability of being a losing image, we observe almost no change in performances. This proves that our method is able to filter out "insufficiently negative samples"–samples that are better than the current model's distribution and subsequently boost performance. Furthermore, we experiment with different sampling strategies for the reference model with AAF. When we set the reference models sampling method to using either the initial checkpoint or the most recent saved checkpoint, we observe a performance drop in both cases. This indicates the effectiveness of our reference model sampling strategy.

Table 3: Metric Scores on the ChronoMagic-Bench-150 Dataset. ↓ indicates the lower the better, and ↑ indicates the higher the better.

|  | FID ↓ | LPIPS ↓ | SSIM ↑ | PSNR ↑ | FVD ↓ |
|---|---|---|---|---|---|
| AnimateDiff | 134.86 | 0.68 | 0.16 | 9.18 | 1608.41 |
| SFT | 129.14 | 0.65 | 0.17 | 9.25 | 1415.68 |
| SePPO | **115.32** | **0.61** | **0.20** | **9.36** | **1300.97** |

Figure 5: Text-to-video generation results of AnimateDiff (Raw), SFT, and SePPO. The prompt is: "*Time-lapse of a lettuce seed germinating and growing into a mature plant. Initially, a seedling emerges from the soil, followed by leaves appearing and growing larger. The plant continues to develop...*"

Table 4: Ablations on the Pick-a-Pic Validation Dataset.

| Methods | PickScore ↑ | HPSv2 ↑ | ImageReward ↑ | Aesthetic ↑ |
|---|---|---|---|---|
| w/o AAF | 20.88 | 26.78 | 0.366 | 5.491 |
| w/ $\mathbb{1}(x > 0)$ | 21.56 | 27.18 | 0.606 | **5.797** |
| Ref as the initial (ref $= 0$) | 21.41 | 27.04 | 0.562 | 5.727 |
| Ref as the latest (ref $= k - 1$) | 21.34 | 27.05 | 0.537 | 5.708 |
| SePPO$^w$ | **21.57** | **27.20** | **0.615** | 5.772 |

# 6 DISCUSSIONS

## 6.1 LIMITATIONS

First, in diffusion models, the theoretical analysis of exploration in policy space constrained by reference models is an open problem. Second, the performance may be further improved if the pixel space (images before encoding) is also considered. We leave this to future work.

## 6.2 CONCLUSION

Without using reward models or human-annotated paired data, we have developed a Semi-Policy Preference Optimization (SePPO) method, which takes previous checkpoints as reference models and uses them to generate on-policy reference samples, which replace "losing images" in preference pairs. In addition, we design a strategy for reference model selection that expands the exploration in the policy space. Furthermore, instead of directly taking reference samples as negative examples, we propose an Anchor-based Adaptive Flipper to determine whether the reference samples are likely to be winning or losing images, which allows the model to selectively learn from the generated reference samples. In text-to-image benchmarks, SePPO achieved a 21.57 PickScore, exceeding all previous approaches on the SD-1.5 model. In addition, SePPO performs better than SFT on text-to-video benchmarks.

## REPRODUCIBILITY STATEMENT

For algorithms, we put the key parts (loss function) in Appendix B.2. We upload the main code for training to the supplementary material. The model checkpoints will be released shortly. For datasets, we use open source datasets described in Section 5.1. For generated results, we upload generated videos to the supplementary material.

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

## A THEORETICAL ANALYSIS

### A.1 PROOF OF THEOREM 4.1

In step $t$, recall that the original image is $\mathbf{x}_0 = \frac{\mathbf{x}_t - \sqrt{1-\alpha^t}\epsilon}{\sqrt{\alpha^t}}$, and the image recovered by the DDPM $\theta$ is defined as $\widehat{\mathbf{x}}_0^\theta := \frac{\mathbf{x}_t - \sqrt{1-\alpha^t}\epsilon_\theta(\mathbf{x}_t)}{\sqrt{\alpha^t}}$. We denote the means of these two Gaussian distributions as $\widehat{\mu}_0^\theta$ and $\mu_0$, respectfully. The performance of a DDPM model $\theta$ can be measured by the KL distance (Chan, 2024) $D_{\mathrm{KL}}(\widehat{\mathbf{x}}_0^\theta || \mathbf{x}_0)$ between the recovered image $\widehat{\mathbf{x}}_0^\theta$ and the original image $\mathbf{x}_0$. Thus, we give the standard of recovery performance from noisy image $\mathbf{x}_t$ in Defination A.1.

**Definition A.1.** Given two DDPMs $\theta_1$ and $\theta_2$, $\theta_1$ is better than $\theta_2$ ($\theta_1 \succ \theta_2$) if its predicted image $\widehat{\mathbf{x}}_0^{\theta_1}$ has less KL divergence with the original image $\mathbf{x}_0$ as follows

$$D_{\mathrm{KL}}(\widehat{\mathbf{x}}_0^{\theta_1} || \mathbf{x}_0) \leq D_{\mathrm{KL}}(\widehat{\mathbf{x}}_0^{\theta_2} || \mathbf{x}_0). \tag{11}$$

In the noise injection process, the variance of the images remains the same. ($\mathbf{x}_0$ and $\mathbf{x}_t$ have the same variance.) With the fact that the KL divergence between two Gaussian distributions with the identical variance is proportional to the Euclidean distance of their means, we have

$$
\begin{aligned}
D_{\mathrm{KL}}(\widehat{\mathbf{x}}_0^\theta || \mathbf{x}_0) &= \frac{1}{2\sigma_0^2} \|\widehat{\mu}_0^\theta - \mu_0\|^2 \\
&= \frac{1}{2\sigma_0^2} \left\| \mathbb{E}\left[\frac{\mathbf{x}_t - \sqrt{1-\alpha^t}\epsilon_\theta(\mathbf{x}_t)}{\sqrt{\alpha^t}}\right] - \mathbb{E}\left[\frac{\mathbf{x}_t - \sqrt{1-\alpha^t}\epsilon}{\sqrt{\alpha^t}}\right] \right\|^2 \\
&= \frac{1-\alpha^t}{2\sigma_0^2\alpha^t} \left\| \mathbb{E}\left[\epsilon_\theta(\mathbf{x}_t)\right] - \mathbb{E}[\epsilon] \right\|^2 \\
&= \frac{1-\alpha^t}{2\sigma_0^2\alpha^t} \mathbb{E}\left[\|\epsilon_\theta(\mathbf{x}_t) - \epsilon\|^2\right] - \frac{1-\alpha^t}{2\sigma_0^2\alpha^t}.
\end{aligned}
\tag{12}
$$

The last step is from Jensen's inequality for the square-error function. Thus, given the condition

$$\mathbb{E}\left[\|\epsilon_{\theta_1}(\mathbf{x}_t) - \epsilon\|^2 - \|\epsilon_{\theta_2}(\mathbf{x}_t) - \epsilon\|^2\right] \leq 0, \tag{13}$$

we have

$$D_{\mathrm{KL}}(\widehat{\mathbf{x}}_0^{\theta_1} || \mathbf{x}_0) - D_{\mathrm{KL}}(\widehat{\mathbf{x}}_0^{\theta_2} || \mathbf{x}_0) \leq 0. \tag{14}$$

The prediction from model $\theta_1$ has a smaller KL distance compared to the prediction from model $\theta_2$. Thus, $\theta_1$ recovers better images and $\theta_1 \succ \theta_2$ by the definition of performance measurement. As a result, $\theta_1$ has a higher probability of generating a good result $\mathbf{x}_0^{\theta_1}$.

## B SUPPLEMENTARY EXPERIMENTS

### B.1 ABLATION STUDY ON ITERATION $K$

In this subsection, we perform an ablation study w.r.t. the number of iterations $K$. In Figure 6, we found that when changing the total number of iteration $K$ for saving checkpoints, relatively, the larger $K$ achieves better performance. However, the overall trend does not change significantly, which demonstrates the stability of SePPO on $K$.

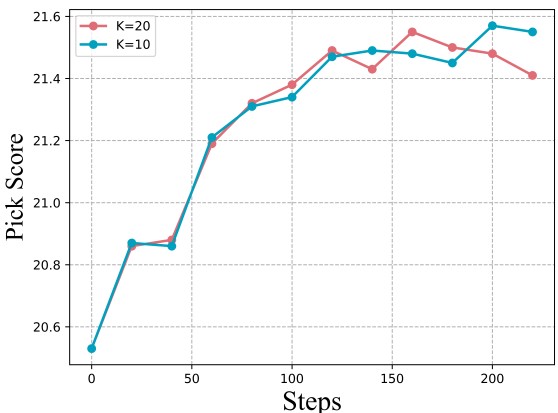

Figure 6: Ablation study of $K = 10$ and $K = 20$.

## B.2 PSEUDOCODE

In this subsection, we give the pseudocode of the loss function for training.

```python
def loss(target, model_pred, ref_pred, scale_term):
    #### START LOSS COMPUTATION ####
    if args.train_method == 'sft': # SFT, casting for F.mse_loss
        loss = F.mse_loss(model_pred.float(), target.float(),
                          reduction="mean")
    elif args.train_method == 'seppo':
        # model_pred and ref_pred will be
        # (2 * LBS) x 4 x latent_spatial_dim x latent_spatial_dim
        # losses are both 2 * LBS
        # 1st half of tensors is preferred (y_w),
        # second half is unpreferred

        model_losses = (model_pred - target).pow(2).mean(dim=[1,2,3])
        model_losses_w, model_losses_l = model_losses.chunk(2)

        with torch.no_grad():
        # Get the reference policy (unet) prediction
            ref_pred = ref_unet(*model_batch_args,
                                added_cond_kwargs=added_cond_kwargs
                                ).sample.detach()

            ref_losses = (ref_pred - target).pow(2).mean(dim=[1,2,3])
            ref_losses_w, ref_losses_l = ref_losses.chunk(2)

        pos_losses_w = model_losses_w - ref_losses_w

        sign = torch.where(pos_losses_w > 0,
                                   torch.tensor(1.0),
                                   torch.tensor(-1.0))

        model_diff = model_losses_w + sign * model_losses_l
        ref_diff = ref_losses_w + sign * ref_losses_l

        scale_term = -0.5 * args.beta_dpo
        inside_term = scale_term * (model_diff - ref_diff)
        implicit_acc = (inside_term > 0).sum().float() /
                        inside_term.size(0)
        loss = -1 * (F.logsigmoid(inside_term)).mean()

    return loss
```

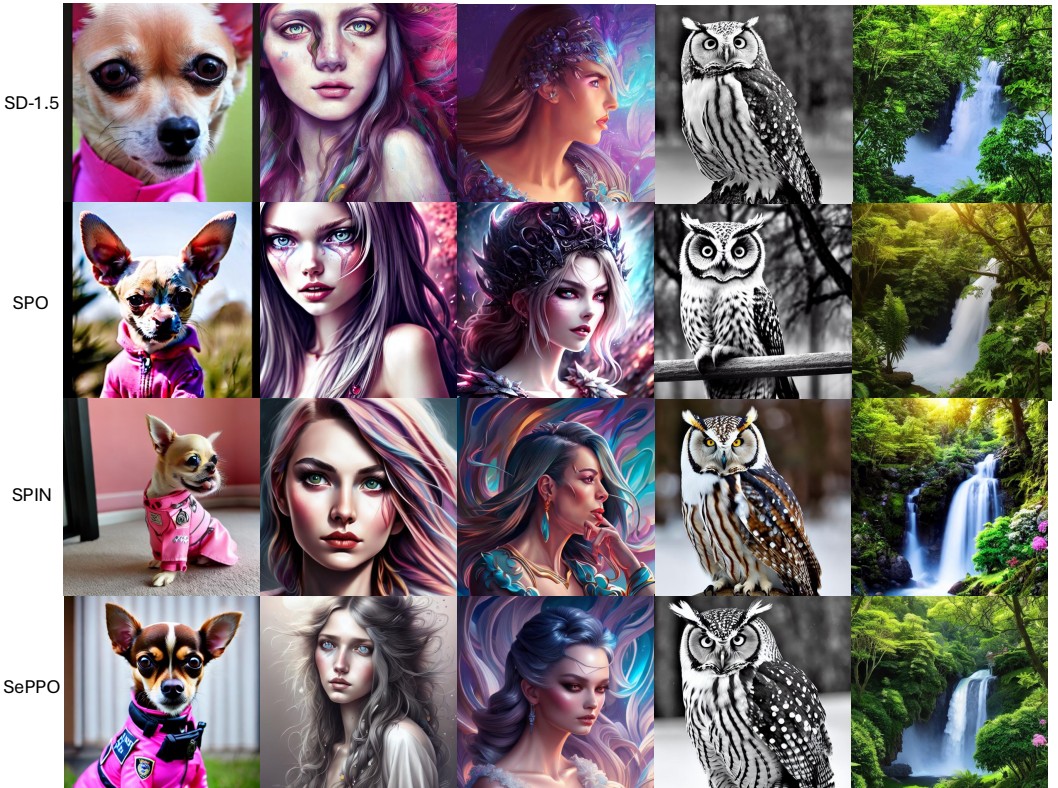

Figure 7: Text-to-image generation results of SD-1.5, SPO, SPIN-Diffusion, and SePPO. Prompts from left to right: (1) *Pink Chihuahua in police suit*; (2) *Detailed Portrait Of A Disheveled Hippie Girl With Bright Gray Eyes By Anna Dittmann, Digital Painting, 120k, Ultra Hd, Hyper Detailed, Complimentary Colors, Wlop, Digital Painting*; (3) *Chic Fantasy Compositions, Ultra Detailed Artistic, Midnight Aura, Night Sky, Dreamy, Glowing, Glamour, Glimmer, Shadows, Oil On Canvas, Brush Strokes, Smooth, Ultra High Definition, 8k, Unreal Engine 5, Ultra Sharp Focus, Art By magali villeneuve, rossdraws, Intricate Artwork Masterpiece, Matte Painting Movie Poster*; (4) *winter owl black and white*; (5) *You are standing at the foot of a lush green hill that stretches up towards the sky. As you look up, you notice a beautiful house perched at the very top, surrounded by vibrant flowers and towering trees. The sun is shining brightly, casting a warm glow over the entire landscape. You can hear the sound of a nearby waterfall and the gentle rustling of leaves as a gentle breeze passes through the trees. The sky is a deep shade of blue, with a few fluffy clouds drifting lazily overhead. As you take in the breathtaking scenery, you can't help but feel a sense of peace and serenity wash over you*.

## B.3 VISUAL GENERATION EXAMPLES

We present more text-to-visual generation results of SePPO and other methods. In Figure 7, we show the text-to-image generation results of SD-1.5, SPO, SPIN-Diffusion, and SePPO. In Figure 8 and Figure 9, we show the text-to-video generation results of AnimateDiff (Raw), SFT, and SePPO.

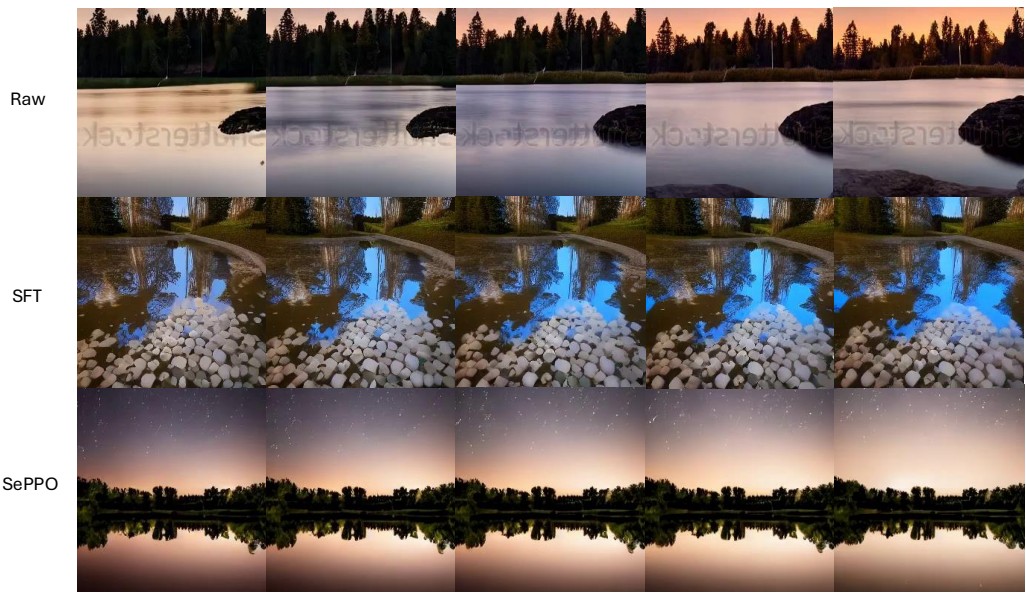

Figure 8: Text-to-video generation results of AnimateDiff (Raw), SFT, and SePPO. Prompt: *"Time-lapse of night transitioning to dawn over a serene landscape with a reflective water body. It begins with a starry night sky and minimal light on the horizon, progressing through increasing light and a glowing horizon, culminating in a serene early morning with a bright sky, faint stars, and clear reflections in the water."*

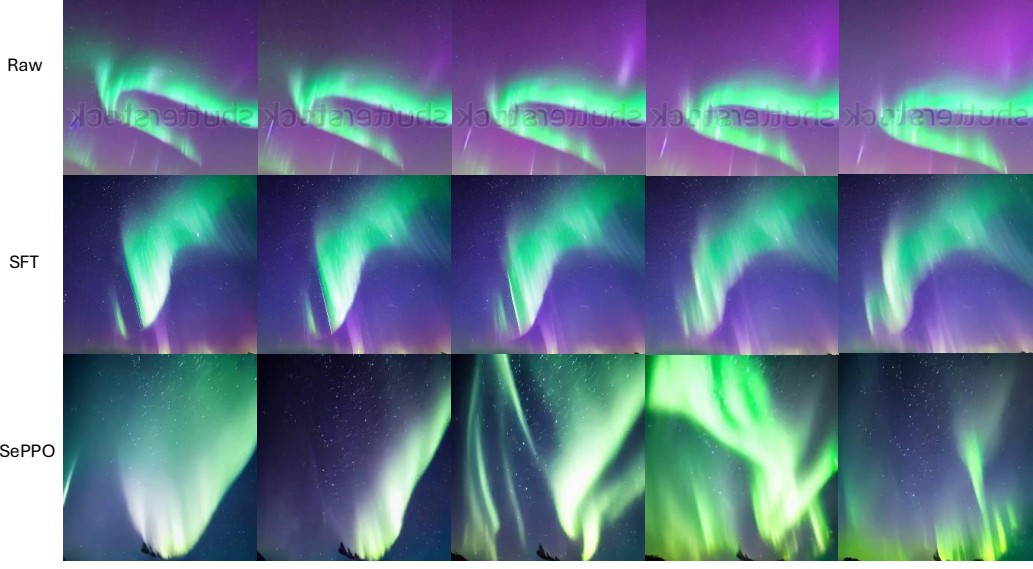

Figure 9: Text-to-video generation results of AnimateDiff (Raw), SFT, and SePPO. Prompt: *"Time-lapse of aurora borealis over a night sky: starting with green arcs, intensifying with pronounced streaks, and evolving into swirling patterns. The aurora peaks in vivid hues before gradually fading into a homogeneous glow on a steadily brightened horizon."*

