# OpenReview forum: "SePPO: Semi-Policy Preference Optimization for Diffusion Alignment"
_ICLR.cc/2025/Conference — ICLR 2025 Conference Withdrawn Submission_

### Official Review · Reviewer_QoUj · 2024-11-02

**Soundness:** 3
**Presentation:** 3
**Contribution:** 2
**Rating:** 5
**Confidence:** 3

**Summary:**

The paper proposes a diffusion model to improve sample quality and diversity in generative tasks. The authors introduce an algorithm that integrates an Anchor-based Adaptive Flipper. To substantiate the claims, a series of comprehensive experiments, including quantitative evaluations against several state-of-the-art models and detailed ablation studies, are presented.

**Strengths:**

1.	This paper employs thorough quantitative metrics such as PickScore and HPSv2. The use of ablation studies is commendable, clearly delineating the contributions of individual components of the proposed model.
2.	The results are well-presented, with clear visualizations and comprehensive tables that facilitate an understanding of performance metrics across different models.
3.	The proposed method is simple and easy to follow.

**Weaknesses:**

1.	The proposed method is a bit tricky, which may limit its contribution. Therefore, I suggest the authors conduct a deeper theoretical analysis of the proposed method.
2.	The insights from Theorem 4.1 are quite intuitive and easy to understand. I suggest the authors put Theorem 4.1 in the Appendix.
3.	The essential reason why randomly sampling previous checkpoints as a reference model with AAF achieves the best performance is still unclear. I suggest the authors theoretically analyze the effectiveness of the proposed method, which could significantly strengthen this work. At least, the authors need to analyze in which cases, using the latest model is better than randomly sampling previous checkpoints as the reference model.

**Questions:**

Please see the Weaknesses.

---

### Official Review · Reviewer_3U9X · 2024-11-02

**Soundness:** 2
**Presentation:** 3
**Contribution:** 2
**Rating:** 5
**Confidence:** 5

**Summary:**

The paper introduces Semi-Policy Preference Optimization for fine-tuning diffusion models in visual generation, bypassing reward models and human annotations. SePPO uses past model checkpoints to generate on-policy reference samples, replacing “losing” images, and focuses on optimizing only "winning" samples. An anchor-based criterion selectively learns from these reference samples, mitigating performance drops from uncertain quality. SePPO outperforms existing methods in text-to-image and shows strong results in text-to-video benchmarks.

**Strengths:**

1. **Practical Approach**: The SePPO method offers a practical solution for preference alignment without the need for human annotation or a reward model, which reduces significant labor costs.
2. **Clear Writing and Presentation**: The submission is well-written and formatted, making it easy to follow and understand.
3. **Effective Sample Filtering**: The Anchor-based Adaptive Flipper (AAF) criterion is a useful addition, as it helps to filter uncertain samples and enhances model robustness.

**Weaknesses:**

1. **Limited Literature Review and Comparison**: While preliminary experiments are presented for text-to-video models, the paper lacks a thorough literature review on this topic and has limited comparisons in the text-to-video experiments, making the evaluation seem somewhat incomplete. Improving the Related Work section would strengthen the context and position of SePPO.
2. **Table Readability**: Moving the annotation explanations to the table captions could improve table readability.
3. **Unclear Justification for Theorem 4.1**: The key of instantiating SePPO is the Theorem 4.1, however, it remains a question to me about this rationality. Specifically, if the reference model has a higher probability of generating noise compared to the current model, then in this situation, we should say this model is a better model for generating this image. We cannot assert the quality of this generated image.

**Questions:**

Generally speaking, methods like Diffusion-DPO requires human-annotated data pairs and SePPO does not. The upper bound of aligning model outputs using annotated data pairs should be higher than SePPO, which solely relies on the model itself. Can the authors present an explanation for this?

---

### Official Review · Reviewer_Yyer · 2024-11-04

**Soundness:** 2
**Presentation:** 3
**Contribution:** 2
**Rating:** 5
**Confidence:** 4

**Summary:**

This paper proposes SePPO, leveraging two techniques to improve the previous SPIN-Diffusion method, including 1) randomly selecting the reference policy from all previous checkpoints and 2) A heuristic (anchor-based criterion) to determine whether a reference sample will likely win or lose. The paper performs experiments on both T2I and T2V tasks to demonstrate the effectiveness of their methods by comparing them with several different methods.

**Strengths:**

- The paper performs experiments on both T2I and T2V generation tasks.

- The paper is easy to follow and understand.

**Weaknesses:**

- **Technical contributions**: The proposed techniques involve 1) randomly selecting the reference policy from the checkpoint at iteration 0 (DITTO) and the latest iteration t - 1 (SPIN-Diffusion). 2) A heuristic (anchor-based criterion) to determine whether a reference sample will likely win or lose, and the learning loss is adjusted accordingly. Thus, the technical contributions contributions of this work are limited.

- **Incorrect Definition of on-policy examples**: The definitions of on-policy and off-policy learning are well-defined in reinforcement learning literature [1]. Specifically, on-policy learning refers to the settings when the training examples are sampled from the current policy ($\pi_\theta$) being learned. However, this work treats the reference samples $\mathbf{x}^{ref}\_0$ sampled from $\pi\_\{ref}$ as "on-policy" examples (e.g., Line 250 - 252), **which is incorrect**. In fact, both $\mathbf{x}^{ref}\_0$ and $\mathbf{x}^w\_0$ are off-policy samples since none of them is sampled from the current policy $\pi_\theta$.

- **Limited Performance Improvement**: According to Tables 1 and 2, the performance improvement of the SePPO$^w$ over SPIN-Diffusion is trivial in terms of PickScore and HPSv2 score. The improvement is only obvious when evaluating with ImageReward. Additionally, SPIN-Diffusion even outperforms the proposed SePPO$^w$ by an obvious margin in terms of Aesthetic score. Therefore, I would recommend conducting a human evaluation to corroborate the results as in [2].

- **Evaluation protocol for video generation tasks**: The metrics used in Table 3 are not meaningful enough. As the model is trained on ChronoMagic-Bench-150, I recommend reporting the results by following the evaluation protocol in Tables 3 & 4 of the ChronoMagic-Bench paper [4].

- **Missing Citations**: Please cite the related works [2] and [3], which tackle T2I and T2V model alignment by learning from reward models.

**Minor points**
1. Line 025: "winning or losing images" --> "winning or losing examples". Since the proposed method is not limited to image generation, please revise similar errors throughout the paper.

2. I recommend not using too many subsubsections. Furthermore, avoid using unnecessary new lines when formalizing the optimization problems (e.g., Equation 5). If you are worried about the page limit, please include more qualitative examples in the main text.

3. I suggest selecting a different abbreviation for your method. PPO [5] in is widely recognized as an algorithm focused on learning a reward function. Since your SePPO is in the self-play finetuning family and is unrelated to PPO, using this acronym may lead to confusion.

[1] Sutton, Richard S. "Reinforcement learning: An introduction." A Bradford Book (2018).

[2] Li et al., "Reward Guided Latent Consistency Distillation", TMLR 2024

[3] Li et al., "T2V-Turbo: Breaking the Quality Bottleneck of Video Consistency Model with Mixed Reward Feedback", NeurIPS 2024

[4] Yuan et al., "ChronoMagic-Bench: A Benchmark for Metamorphic Evaluation of Text-to-Time-lapse Video Generation". NeurIPS 2024

[5] Schulman et al., "Proximal Policy Optimization Algorithms".

**Questions:**

Please refer to the Weakness section.

---

### Official Review · Reviewer_oHX4 · 2024-11-04

**Soundness:** 2
**Presentation:** 2
**Contribution:** 2
**Rating:** 5
**Confidence:** 3

**Summary:**

This paper proposes SePPO, a novel preference optimization method for aligning diffusion models with human preferences without requiring reward models or paired human-annotated data. The key innovations are: 1) Using previous checkpoints as reference models to generate on-policy reference samples, 2) Introducing a strategy for reference model selection that enhances policy space exploration, and 3) Developing an Anchor-based Adaptive Flipper (AAF) to assess reference sample quality. The method shows strong performance on both text-to-image and text-to-video generation tasks, outperforming previous approaches across multiple metrics.

**Strengths:**

1. The paper proposes a combination of semi-policy optimization with the AAF mechanism without requiring reward models or paired human-annotated data.
2. Comprehensive Empirical Validation: The work provides comprehensive experimental validation across both text-to-image and text-to-video domains.

**Weaknesses:**

1. I find the definition of "better" (L277 in bold) to be confusing and the same term shows up in Theorem 4.1 seems lacking rigor. I think what the authors mean is that "closer" to the preferred sample $x_0^w$, but closer to $x_0^w$ does not necessarily mean better since it depends on the metric considered. Given a reward function where $r(x_0^w)>r(x_0^l)$, whether a new sample $x_1, x_2$ has a higher reward is dependent on the reward landscape, not how close it is to $x_0^w$.
2. Given 1, I think the main spirit of the proposed method is to fit the preferred distribution, similar to SPIN. In that sense, I am confused about why the proposed method is expected to do better since the advantage of the proposed method compared to SPIN is not clearly discussed in the paper. For example, what is missing in SPIN that the proposed method can do?
3. Lack of human evaluation.

**Questions:**

See weaknesses.

---

### Author Response · Authors · 2024-11-15
**Thanks for the valuable suggestions!**

We would like to appreciate the reviewers' efforts and the valuable suggestions. We will improve the paper and **address all issues** accordingly in the next submission.

---

### Note · Authors · 2024-11-15

I have read and agree with the venue's withdrawal policy on behalf of myself and my co-authors.